# Impossibility Results for Grammar-Compressed Linear Algebra

**Amir Abboud**
IBM Almaden Research Center
amir.abboud@gmail.com

**Arturs Backurs**
Toyota Technological Institute at Chicago
backurs@ttic.edu

**Karl Bringmann**
Saarland University and
Max Planck Institute for Informatics,
Saarland Informatics Campus (SIC)
bringmann@cs.uni-saarland.de

**Marvin Künnemann**
Max Planck Institute for Informatics,
Saarland Informatics Campus (SIC)
marvin@mpi-inf.mpg.de

## Abstract

To handle vast amounts of data, it is natural and popular to compress vectors and matrices. When we compress a vector from size $N$ down to size $n \ll N$, it certainly makes it easier to store and transmit efficiently, but does it also make it easier to process?

In this paper we consider lossless compression schemes, and ask if we can run our computations on the compressed data as efficiently as if the original data was that small. That is, if an operation has time complexity $T(\text{input-size})$, can we perform it on the compressed representation in time $T(n)$ rather than $T(N)$? We consider the most basic linear algebra operations: inner product, matrix-vector multiplication, and matrix multiplication. In particular, given two compressed vectors, can we compute their inner product in time $O(n)$? Or perhaps we must decompress first and then multiply, spending $\Omega(N)$ time?

The answer depends on the compression scheme. While for simple ones such as Run-Length-Encoding (RLE) the inner product can be done in $O(n)$ time, we prove that this is impossible for compressions from a richer class: essentially $n^2$ or even larger runtimes are needed in the worst case (under complexity assumptions). This is the class of *grammar-compressions* containing most popular methods such as the Lempel-Ziv family. These schemes are more compressing than the simple RLE, but alas, we prove that performing computations on them is much harder.

## 1 Introduction

The idea of using compression to speed up computations can be found in any domain that deals with large-scale data, and ML is no exception. By exploiting redundancies and various forms of structure in a piece of data, compression algorithms such as zip can reduce its size from $N$ down to $n$, where $n \ll N$. The data becomes cheaper to store, access, transmit, and perhaps also to analyze. Can we run our ML tools on the compressed data, without decompressing it first, and make the computation times proportional to $n$ rather than $N$? Since most ML algorithms boil down to large amounts of basic algebraic operations such as multiplications of vectors and matrices, with *inner product* as the atomic operation, the most basic question in this context is:

*Main Question.* Given two $N$-dimensional vectors, each in a compressed form of size $n \ll N$, can we compute their inner product in $\tilde{O}(n)$ time[1] rather than $O(N)$?

The answer, of course, depends on the compression scheme that we use. There seems to be an inherent tension: more complex schemes have higher compression rates but are harder to analyze without decompression.

First, let us clarify that our interest is in exact computations and *lossless* compressions, even though lossy techniques such as dimensionality reduction [16] are widely used by the ML community. In many cases, e.g. when performing a basic algebraic operation within a larger pipeline, even a small amount of error could add up to make the final result unintelligible. Recent years has seen a growing interest in exploring the potential of lossless compression for speeding up ML [35, 83, 59, 65]. An inspiring result was honorably mentioned as an outstanding paper at NeurIPS last year [65]: any $N \times d$ matrix $A$ can be compressed down to a matrix of size $d \times d$ such that the optimal solutions of Least-Mean-Squares (LMS) instances are exactly the same on $A$ and $A'$. This is an example where for a specific task (LMS solvers) a specific compression scheme (designed by the authors) leads to a solution in time $T(n)$ rather than $T(N)$, giving a 100x speedup on benchmark data; it makes one wonder if this approach can work in a more general setting.

For rather simple compression methods, the answer to our question is positive. A recent Communications of the ACM article [35] exhibits *Compressed Linear Algebra* [32, 33, 34] a compression scheme for vectors and matrices that uses simple techniques such as Run Length Encoding (RLE) and allows for fast computations on the compressed data with impressive experimental results when integrated into ML systems. The RLE encoding of a vector simply replaces runs of values by tuples indicating the value and the length of the run; e.g. the binary vector 00011111000 gets encoded as $0^3 1^5 0^3$. Given two vectors encoded in this way with size $n_{RLE}$, a simple one-pass algorithm can compute their inner product in $O(n_{RLE})$ time. Before that, there were many algorithms for exploiting succinct encodings of *sparse* vectors [78, 56, 52]; e.g. by simply listing the nonzero locations the binary vector 0100001000 gets encoded as $(2, 7)$. These encodings allow for a linear time inner product computation as well.

However, these simple methods are often not very compressing. At the other end of the spectrum, we have the heavy-duty and time-tested family of *Grammar-Compressions* [54] that includes the Lempel-Ziv-family (LZ77, LZ78, LZW, etc.) [58, 91, 86], Byte-Pair Encoding [82], dictionary methods, and others [69, 63]. These compressions are used in ubiquitous applications such as zip, Snappy, GIF, PNG, the built-in Unix utility `compress`, and even in PDF. Their compression rates are often on a whole different level compared to RLE; e.g. the current draft of this paper reduces from 10KB to 4KB with zip but RLE has no effect. See Table 1 and [35, Table 1] for empirical data showing the quantitative potential of these methods for some standard ML datasets. What all these more elaborate compression techniques have in common is that they essentially (up to low order terms [76]) encode a string (or vector) by a *Straight-Line Program* (SLP): a restricted kind of a context-free grammar that can only produce one string. In more detail, an SLP is defined over some alphabet $\Sigma$, say $\{0, 1\}$, and it is a set of replacement rules (or productions) of a very simple form: a rule is either a symbol in $\Sigma$ or it is the concatenation of two previous rules (under some fixed ordering of the rules). The last replacement rule is the sequence defined by the SLP. For example, we can compress the sequence 01010101 with the rules $S_1 \to 0$; $S_2 \to 1$; $S_3 \to S_1 S_2$; $S_4 \to S_3 S_3$; $S_5 \to S_4 S_4$ and $S_5$ corresponds to the sequence 01010101. See Figure 1. For some strings this can give an exponential compression, e.g. the sequence $(01)^N$ requires only $O(\log N)$ rules; note that its RLE has size $N$. While finding the smallest SLP for a given string is NP-Hard, it can be approximated either by the above practical methods or provably up to logarithmic factors [76, 20, 79, 48, 50].

Thus, the holy grail in this context is to perform algebraic operations in $T(\text{compression-size})$ time *even when* the vectors are compressed with zip or one of the other heavy-duty grammar compressions; that is, without unzipping them first. Ideally, we would implement a "zip-inner-product" function that takes two zip files encoding vectors and computes the inner product in near-linear time (which may not even be enough time to unzip them). A recent paper titled "When LZW meets ML" [59] makes partial progress towards this goal: the inner product can be computed efficiently on their *tuple oriented coding* where each *coordinate* is grammar-compressed separately, but not the

Table 1: The potential savings from grammar-compressed linear algebra: Compression rates on real datasets. We compare zip, a standard grammar-compression, with Run Length Encoding (RLE), a simple method that works well on repetitive or sparse data. For more such results, see [35, Table 1].

| Dataset | Size | RLE (compression rate) | zip (compression rate) |
|---|---|---|---|
| ISOLET [30] | 30.94 MB | 29.83 MB (0.96) | 7.94 MB (0.26) |
| US Census 1990 [30] | 342.26 MB | 341.97 MB (0.99) | 51.91 MB (0.15) |

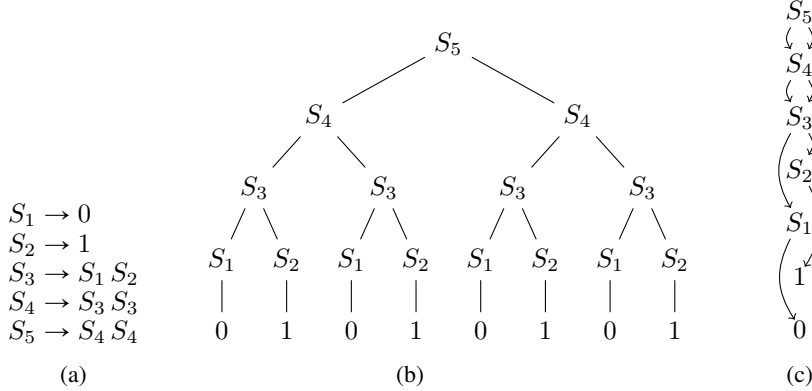

$$S_1 \to 0$$
$$S_2 \to 1$$
$$S_3 \to S_1 S_2$$
$$S_4 \to S_3 S_3$$
$$S_5 \to S_4 S_4$$

(a)                       (b)                       (c)

Figure 1: (a) An SLP generating the sequence $01010101$. (b) The corresponding parse tree. (c) The acyclic graph corresponding to the SLP.

vector as a whole. This makes their method less compressing since, unlike with zip, the size of the encoding is always at least the dimensionality of the vectors.

*Main Question* (Restated). Given two $N$-dimensional vectors, each grammar-compressed down to size $n \ll N$, can we compute their inner product in $\tilde{O}(n)$ time rather than $O(N)$?

While efficiently analyzing these grammars may seem like a daunting task, a large body of works over the last three decades has equipped us with an ingenious toolbox exactly for this purpose. It turns out that many important problems can indeed be solved surprisingly faster than the decompress-then-solve bound, e.g. in pattern matching [71, 53, 11, 36, 18, 61, 40, 45, 49]. This gives hope for a positive answer to our question and that many ML computations could be sped up by operating on grammar-compressions. These algorithms typically look at the parse trees that have $N$ leaves but only $n$ distinctly labelled internal nodes (see Figure 1), and traverse them starting from the root down, while attempting to only spend time proportional to the depth of the tree per distinct label. Using tricks that restructure the grammar to make the tree balanced, the depth can be upper bounded by $O(\log N)$, making the total time $O(n \log N)$. To learn more about this subfield of Algorithm Design, we refer the reader to the surveys [90, 57, 39, 81, 41, 73, 77, 64, 80].

## 1.1 Our Results

Alas, our main result is a negative resolution to the main question above. We apply the tools of theoretical computer science, and the recently blossoming field of *fine-grained complexity*, in order to shed light into the mathematical foundations of Compressed Linear Algebra. We prove new hardness reductions showing cases where the time to compute the inner product must be large (under popular complexity assumptions) even when the vectors have very small grammar compressions. For example, there are $N$-dimensional vectors with grammar-compressions of size $n = O(N^{1/3})$ where the inner product must take $\tilde{\Omega}(n^2)$ time[2] to compute. The consequences to other settings such as matrix-vector multiplication are further explained below. This creates a strong separation between grammar-compressions, where we prove an $\tilde{\Omega}(n^2)$ lower bound, and RLE, where an $O(n)$

algorithm exists. This formally justifies the use of simpler methods in ML systems and guides researchers away from searching for an ultra-efficient "zip-inner-product" function.

**Fine-Grained Complexity**   Negative results are paramount to the success of any scientific discipline. The most prominent framework for proving such results in computer science is the theory of NP-Hardness, where one proves that a problem cannot be solved in polynomial time unless $P = NP$ which would imply breakthrough algorithms for famously-hard problems such as SAT and Subset Sum. Without this theory, countless hours would have been wasted by algorithm designers trying to come up with provable, worst-case, polynomial time algorithms for NP-Hard problems. Due to the increase in data sizes of recent years, the ethos of this theory that "efficient = polynomial" has become obsolete, and a more demanding attitude where "efficient = linear" has arisen. By replacing the polynomial reductions of NP-Hardness with more efficient ones (often linear), fine-grained complexity can prove hardness results even for problems that have polynomial time algorithms. Exemplary results show that linear or subquadratic algorithms for certain problems, which admit quadratic-time algorithms, would refute popular assumptions (conjectures that are similar to but stronger than $P \neq NP$) and have breakthrough consequences for famously hard problems. One of the central assumptions in this theory and in this paper is the 3SUM Conjecture: "*No algorithm can decide, in subquadratic $O(n^{2-\varepsilon})$ time, if there are three numbers that sum to zero among a given set of $n$ numbers*". A recent survey on fine-grained complexity [89] cites dozens of papers, mainly in computational geometry [38] but also in other fields [72, 85, 7, 8, 21, 55, 10, 43], that prove 3SUM-Hardness results showing that their algorithms are optimal up to a refutation of this conjecture. In this paper, we prove the first 3SUM-Hardness results in ML[3] as far as we are aware. The 3SUM assumption and its generalizations that we use in the theorems below are formally defined and discussed in Section 2.

**Vector Inner Product**   Our first and main result is a reduction from 3SUM to compressed inner product of two vectors, negatively resolving our main question.

**Theorem 1.1.** *Assuming the 3SUM conjecture, the inner product of two $N$-dimensional vectors that are grammar-compressed to size $n = \Theta(N^{\frac{1}{4}})$ cannot be computed in $O(n^{2-\varepsilon})$ time where $\varepsilon > 0$.*

Moreover, we strengthen and generalize this result in several ways. First, we address the dependence between $n$ and $N$: could it be that for more or less compressed vectors the picture is different? Using a stronger variant of the 3SUM conjecture, the same lower bound of $n^2$ holds even when $n = N^{1/3}$, and therefore our result can be stated as an $\tilde{\Omega}(N^{\frac{2}{3}})$ lower bound which is quite close to the trivial upper bound of $O(N)$. Moreover, by a (highly nontrivial) boosting of our reduction, in Section 3 we establish an $\tilde{\Omega}(N^{\frac{1}{3}})$ lower bound with $n = N^{\varepsilon}$ for any $\varepsilon \leqslant 1/3$. That is, when the vectors are highly compressed even $n^{10}$ time is not sufficient[4]; this is in stark contrast to the case of RLE-compressed vectors where $O(n)$ is always possible.

**Matrix-Vector Multiplication**   Next, we consider the problem of computing the $M \cdot v$ product of an $N$-dimensional vector $v$ that is compressed to size $n$ with an $N \times N$ matrix $M$ where each row is compressed to size $O(n)$. Perhaps computing these $N$ inner products as a batch can be done faster than computing each separately. Alas, by another significant boosting of our reduction we prove that this is also impossible. While if the encoding is with RLE the product can be computed in $O(Nn)$ time, which is linear in the representation size of the matrix and thus optimal, it turns out that for grammar compressions $\tilde{\Omega}(Nn^2)$ is required. The proof is in Section 4.

**Theorem 1.2.** *Assuming the 3SUM conjecture, the product of an $N \times N$-dimensional matrix, where each row is grammar-compressed to size $n = \Theta(N^{\frac{1}{5}})$, with an $N$-dimensional vector that is grammar-compressed to size $n$ cannot be computed in $O(Nn^{2-\varepsilon})$ time where $\varepsilon > 0$.*

**Matrix Multiplication**  Finally, we consider matrix multiplication of compressed matrices $C = A \cdot B$. There are multiple ways to compress an $N \times N$ matrix: we might compress each row or each column, so that the compression size is $N \cdot n$, or treat the whole matrix as an $N^2$-dimensional vector and compress it to size $n$. Each way may lead to a different time complexity, but no matter which way we choose, the first question to ask, and that will determine the time we can hope for, is: what is the output size? The naïve answer is that the matrix $C$ has size $N \times N$, but since $A$ and $B$ are compressed, shouldn't we expect $C$ to also be representable with a small grammar of size $n \ll N^2$? Unlike the above questions that deal with computation time, this is an information-theoretic question, and in Section 5 we give strong and unconditional negative answers: the matrix $C$ cannot be grammar-compressed to size $o(N^2/\log^2 N)$ *even* when $A$ and $B$ are strongly compressible. Moreover, some of our results hold even when $A$ and $B$ have very small RLE encodings. Therefore, our results should be of interest to the compressed linear algebra project beyond grammar-compressions.

**Technical Remarks**  While the tools for proving NP-Hardness results for grammar-compressed data are old [64], they only apply in the unrealistic setting where $n = \log N$, and we are interested in more fine-grained results. Only recently, a FOCS paper by the authors [2] introduced the techniques for proving such lower bounds. This previous work focused on combinatorial pattern matching problems and the current work extends it to the setting of linear algebra. Our results establish the hardness even of the simplest setting of binary vectors and matrices over $\{0, 1\}$. This setting is particularly studied due to its connection to graphs, where grammar compressions have also received a lot of attention [66, 67]. Moreover, we show that even deciding if the inner product is $0$ or $\geqslant 1$ is hard, and so our lower bounds hold against any bounded approximation algorithms. Extending the lower bounds to other functions such as computing the $\ell_2$ distance between two vectors is also easy. Like almost all results in fine-grained complexity [89], our lower bounds are against both deterministic and randomized algorithms.

Finally, we remark that our lower bounds are for the most basic setting of *worst-case* instances. Extending them to *average-case* results, showing that instances that come from certain natural distributions are also hard, is an open question. However, notice that even if the original vectors come from a natural distribution, the distribution of the grammar representations will be completely different (and probably far from natural). Therefore, exploiting the structure of non-worst-case instances seems far beyond current reach in this context.

## 1.2   Other Related Works

There have been a few recent works showing fine-grained complexity results for machine learning problems. In particular, [14] showed that the classic algorithm of Viterbi that computes the most likely path in a Hidden Markov Model which results in a given sequence of observations is essentially optimal assuming certain complexity theoretical hypotheses. Another work [13] showed conditional hardness results for multiple empirical risk minimization problems such as kernel support vector machines, kernel ridge regression, and training the final layer of a neural network. Furthermore, there are many works that show hardness for problems that are used in machine learning literature. This includes conditional lower bounds for kernel low-rank approximation [68], closest pair and its variants [9, 75, 88, 24, 29, 28], maximum inner product [6, 22, 23], earth mover's distance (a.k.a. Wasserstein metric) [74], dynamic time warping distance [3, 17].

Further contexts in which lossless compressions are used for ML applications, where the primary focus is on other aspects than increasing algorithmic performance, include compressing and accelerating models for deployment on resource-constrained devices (see [44, 26]; e.g., lossless compressions are used to compress weights after a quantization step) or implementing the principle of minimum description length for feature learning (see [70]).

Outside of ML, the idea of improving efficiency by operating on (losslessly) compressed data is well-established in databases [1, 25, 87, 46], and is gaining traction also in bioinformatics [84].

## 2   Preliminaries

As described in Section 1, a grammar compression of a sequence (or a vector) is an SLP that produces the sequence. In our proofs we will use the following simple observation about SLPs.

**Proposition 2.1.** *Let $\mathcal{G}$ be an SLP with start symbol $S$ that generates a sequence $s$. For any $\alpha \in \mathbb{N}$, we can compute an SLP $\mathcal{G}'$ that generates the $\alpha$-fold repetition of $s$, i.e.,*

$$s^\alpha = \underbrace{s\; s\; \cdots\; s}_{\alpha\ times},$$

*and has size $|\mathcal{G}| + O(\log \alpha)$ in time $O(|\mathcal{G}'|)$.*

*Proof sketch.* Using $O(\log \alpha)$ repeated squaring rules $S_i \to S_{i-1} S_{i-1}$ and $S_0 \to S$, we obtain non-terminals $S_0, \ldots, S_{\lfloor \log_2 \alpha \rfloor}$ generating $s^{2^i}$ for $i \in \{0, \ldots, \lfloor \log_2 \alpha \rfloor\}$. It is straightforward to combine these non-terminals, according to the binary representation of $\alpha$, to generate $s^\alpha$ using only $O(\log \alpha)$ additional non-terminals. $\qquad\square$

Using this property, we can often compress sequences much more efficiently than run-length encoding alone could: E.g., repetitive patterns like $(010011)^n$ can be encoded using only $\Theta(\log n)$ bits instead of $\Theta(n)$. Indeed, our constructions crucially exploit a repeated application of this property to compress hard instances to very small sizes.

**The Complexity Assumptions** As discussed in Section 1, the impossibility results in fine-grained complexity are based on certain popular conjectures. One of the central ones concerns the 3SUM problem, which has a few equivalent formulations (up to linear time transformations [31]); we will mostly use the following[5].

**Definition 2.2** (The 3SUM Problem). *Given three sets $A, B, C$ of $m$ integers in $\{1, \ldots, U\}$, decide if there is a triple $a \in A, b \in B, c \in C$ such that $a + b = c$.*

It is a simple exercise (that is often given in interviews) to come up with an $O(m^2)$ time algorithm, and despite decades of efforts, only mildly subquadratic $O(m^2/\log^c m)$ bounds for a small $0 < c < 3$ are known [15, 51, 42, 37, 19].

*The 3SUM Conjecture.* No algorithm can solve the 3SUM problem in $O(m^{2-\varepsilon})$ time, where $\varepsilon > 0$.

A few remarks about this conjecture. First, a folklore trick of taking all numbers modulo a random large prime shows that the problem for arbitrary universe $U$ is equivalent to the case where $U = O(m^3 \log^2 m)$ (see Lemma B.1 in [5] for a proof). Therefore, we will assume this bound on $U$. When $U$ becomes too small, the problem becomes easy due to an $O(m + U \log U)$ algorithm using Fast Fourier Transform [27]. However, the problem is conjectured to be hard even when $U = \Theta(m^2)$ and this is referred to as the Strong 3SUM Conjecture [10, 2]. This stronger assumption allows us to strengthen our lower bounds by reducing $N$. Second, the hardness of the more general kSUM problem is also used as a complexity assumption [4, 2]. In the formulation that we will use, we are given $k$ sets $A_1, \ldots, A_k$ of $m$ integers in $\{1, \ldots, U\}$ where $U = \Theta(m^{\lceil k/2 \rceil})$ and are asked to decide if there are $k$ numbers, one from each set, such that $a_1 + \cdots + a_{k-1} = a_k$. The Strong kSUM conjecture states that cannot be done in $O(m^{\lceil k/2 \rceil - \varepsilon})$ time, for any $\varepsilon > 0$. We will use this assumption to prove lower bounds even when $n$ is much smaller than $N$. Third, 3SUM and the other hardness assumptions in fine-grained complexity are conjectured to be true even against randomized algorithms that succeed with high probability. This is important since some of our reductions are randomized.

## 3 Vector Inner Product

In this section we present the proof of Theorem 1.1 by giving a reduction from 3SUM to the inner product of compressed vectors. A slightly weaker conditional lower bound of $\tilde{\Omega}(N^{1/2})$ for vectors compressible to $n = N^{1/4}$ can be extracted from the proof of Theorem 5.11 in [2]. We use similar tricks, but a different and more optimized construction to obtain a stronger conditional lower bound of $\tilde{\Omega}(N^{2/3})$ already on less compressible vectors with $n = N^{1/3}$. Technically, the novelty is that we manage to encode two sets ($A$ and $B$) into one vector of length $mU$ rather than $m^2 U$. This new construction is crucial for the extensions we show – we do not see how to prove any lower bound for matrix-vector inner product without building on this new construction.

*Proof.* Given an instance of 3SUM, that is, three sets $A, B, C$ of $m$ integers in $\{1, \ldots, U\}$, we show how to construct vectors $v'_{A+B}, v'_C \in \{0,1\}^N$ with $N = 2mU \log^2 m$ such that: (1) $v'_{A+B} \cdot v'_C \geqslant 1$ if and only there $a \in A, b \in B, c \in C$ with $a + b = c$, (2) both vectors have a compression of size $O(m \log U)$, and (3) the construction time is $O(m \log U)$.

This reduction suffices for proving Theorem 1.1 due to the following calculations. Since (as discussed in Section 2) we can assume that $U = \Theta(m^3 \log^2 m)$, the reduction produces two vectors of dimension $N = \Theta((m \log m)^4)$ and compressed size $n = \Theta(N^{1/4}) = \Theta(m \log m)$, such that the inner product reveals the answer to the 3SUM instance. Therefore, an $O(n^{2-\varepsilon})$-time algorithm would solve the 3SUM instance in time $O(m^{2-\varepsilon} \text{polylog} m)$, refuting the 3SUM conjecture. Note that the $O(m \log U)$ time for the reduction itself is negligible. Moreover, if we assume the Strong 3SUM conjecture, we can start with 3SUM instances where $U = O(m^2)$ and get vectors of dimension $N = O((m \log m)^3)$, ruling out inner product algorithms with time $O(N^{\frac{2}{3}-\varepsilon})$.

We now present the construction of the vectors. As a first step, we observe that for any set $X \subseteq \{1, ..., U\}$, we can compress its characteristic vector $v_X \in \{0,1\}^U$, i.e., $v_X[i] = 1$ iff $i \in X$, to size $O(|X| \log U)$ as follows. We write $X = \{x_1, \ldots, x_{|X|}\}$ with $x_1 < x_2 < \cdots < x_{|X|}$ and observe that
$$v_X := 0^{x_1 - 1} 1 0^{x_2 - x_1 - 1} 1 \ldots 1 0^{x_{|X|} - x_{|X|-1} - 1} 1 0^{U - x_{|X|}},$$
where each 0-block has length at most $U$ and can thus be encoded using $O(\log U)$ symbols using Proposition 2.1. In total, we obtain a compression of size $O(|X| \log U)$, which can be computed in time $O(|X| \log U)$ as well.

Let $A = \{a_1, \ldots, a_n\}$. The central idea is to let $v'_{A+B}, v'_C$ consist of $m$ blocks of size $2U$, where the $i$-th block in $v'_{A+B}$ gives the characteristic vector of the set $a_i + B = \{a_i + b \mid b \in B\} \subseteq \{1, \ldots, 2U\}$ and the $i$-th block in $v'_C$ gives the characteristic vector of $C \subseteq \{1, \ldots, 2U\}$. Formally, we define

$$
\begin{aligned}
v'_{A+B} &:= \underbrace{0^{a_1} v_B 0^{U-a_1}}_{v'_{a_1+B}} \quad \underbrace{0^{a_2} v_B 0^{U-a_2}}_{v'_{a_2+B}} \quad \ldots \quad \underbrace{0^{a_m} v_B 0^{U-a_m}}_{v'_{a_m+B}} \quad 0^{N-2mU}, \\
v'_C &:= \quad v_C 0^U \qquad\quad v_C 0^U \qquad \ldots \qquad v_C 0^U \qquad\quad 0^{N-2mU}.
\end{aligned}
$$

(Here, the last block of 0s only serves to get the desired dimension of $N$ for technical reasons.) We observe that $v'_{A+B}$ and $v'_C$ have an inner product of at least 1 if and only if the characteristic vectors of some block $i$ have a common 1-entry. Thus, consider any block $i$: We have $v'_{a_i+B}[k] = (v_C 0^U)[k] = 1$ if and only if $k - a_i \in B$ and $k \in C$, i.e., $a_i \in A, k - a_i \in B, k \in C$ is a solution of the given 3SUM instance. Thus, $v'_{A+B} \cdot v'_C \geqslant 1$ if and only if there is some $a \in A, b \in B, c \in C$ such that $a + b = c$, as desired.

It remains to show that a $O(m \log U)$-sized compression of $v'_{A+B}$ and $v'_C$ can be computed in time $O(m \log U)$: Clearly, since $v_C 0^U$ can be compressed to size $O(m \log U)$ efficiently, we can also compress its $m$-fold repetition using $O(\log m)$ additional symbols using Proposition 2.1, as well $0^{N-2mU}$ which takes $O(\log N) = O(\log mU)$ additional symbols; thus, $v'_C$ can be compressed to size $O(m \log mU)$ in time $O(m \log U)$. Furthermore, recall that we can compress $v_B$ to size $O(m \log U)$ efficiently, and let $\mathcal{G}$ be an SLP with starting symbol $S_B$ generating $v_B$. Thus, to compress $v_{a_i+B}$, we only need to compress the surrounding blocks $0^{a_i}, 0^{U-a_i}$ and can reuse $S_B$ to generate $v_B$. Since we can encode the 0-blocks using $O(\log U)$ additional non-terminals, this yields a compression size of $O(\log U)$ per block $i$. Together with a $O(\log mU)$ encoding of the trailing block $0^{N-2mU}$, this yields again a compression of size $O(m \log U)$. Note that reusing a non-terminal generating $v_B$ was instrumental in giving a compression of size $O(m \log m)$ rather than $O(m^2 \log m)$ and that this compression can indeed be computed in time $O(m \log U)$ and concludes the claim. $\qquad \square$

With more work, the above arguments can be generalized to reduce a $k$SUM instance with $k$ sets of $m$ integers in $\{1, \ldots, U\}$ to vectors of dimension $N = \Theta(m^{k-2}U)$ and compressed size $O(m \log U)$ in time $O(m \log U)$. The main idea is to encode a shift of $A_{k-1}$ for each tuple of $A_1, \ldots, A_{k-2}$ in one vector, and encode $m^{k-2}$ repetitions of the remaining set $A_k$ in the other vector. Under the Strong $k$SUM conjecture, this yields a conditional lower bound for inner product of $\tilde{\Omega}(N^{1/3})$ where $n = O((N/U)^{1/(k-2)} \log N)$. Thus, for any fixed $\varepsilon > 0$, let $k$ be a sufficiently large constant integer

such that $1/(k-2) < \varepsilon$, then the Strong $k$SUM conjecture implies that $N$-dimensional vectors with compressed size $n = O(N^\varepsilon)$ cannot have an $O(N^{1/3-\delta})$ algorithm for any constant $\delta > 0$. We formally prove the result in the appendix.

# 4 Matrix-Vector Multiplication

In this section we sketch how to prove Theorem 1.2 by giving a reduction from 3SUM to Matrix-Vector multiplication on compressed data. We give a complete formal proof in the appendix.

A helpful tool for this task is the following self-reduction for 3SUM, which follows from combining a known self-reduction [62] with a standard universe-size reduction technique on each produced instance [15, 72, 5].

**Lemma 4.1** (Self-Reduction for 3SUM). *Let* $1 \leqslant s = s(m) \leqslant m$ *and* $\epsilon > 0$ *be arbitrary. If there is an algorithm that, given a target* $t$ *and* $L = O((m/s)^2)$ *sets* $A_\ell, B_\ell, C_\ell$ *of* $s$ *integers in* $\{1, \ldots, O(s^3 \log^2 s)\}$, *determines for all* $1 \leqslant \ell \leqslant L$ *whether there are* $a \in A_\ell, b \in B_\ell, c \in C_\ell$ *with* $a + b + c = t$ *in total time* $O(m^{2-\epsilon})$, *then the 3SUM conjecture is false.*

Given the above self-reduction, the basic idea is as follows. We construct a matrix $M$ whose rows are indexed by the instance $1 \leqslant \ell \leqslant L$ and the aim is to construct the row $M_\ell$ and the vector $v$ such that $M_\ell \cdot v \geqslant 1$ if and only if the instance $A_\ell, B_\ell, C_\ell$ contains a solution, i.e., $a \in A_\ell, b \in B_\ell, c \in C_\ell$ with $a + b + c = t$. Unfortunately, we cannot apply our Vector Inner Product construction directly: this would encode the set $A_\ell + B_\ell = \{a + b \mid a \in A_\ell, b \in B_\ell\}$ into the row $M_\ell$ and the set $C_\ell$ into the vector $v$ – however, in the matrix product $Mv$, each row $M_\ell$ is multiplied with a *fixed* vector $v$, while the $C_\ell$'s differ for each $\ell$. We overcome this issue by adapting our construction to encode the set $A_\ell + B_\ell + C_\ell = \{a + b + c \mid a \in A_\ell, b \in B_\ell, c \in C_\ell\}$ into the row $M_\ell$, and only the common target $t$ into $v$. As all instances use the same target $t$, this is indeed possible.

Specifically, using the ideas of Theorem 1.1, which produces a $2sU$-dimensional vectors encoding the sets $A + B$ and $C$, both having compressed size $O(s \log U)$, we show how to produce $3s^2U$-dimensional vectors $M_\ell$ and $v$ encoding the sets $A_\ell + B_\ell + C_\ell$ and $\{t\}$, both having compressed size $O(s \log U)$. This yields a $(L \times 3s^2U)$-dimensional matrix $M$ and $3s^2U$-dimensional vector $v$. There is a choice $s = \Theta(m^{2/7})$ that leads to a quadratic matrix $M$ with dimension $N = \Theta(m^{10/7})$ (as it has $O((m/s)^2) = O(m^{10/7})$ rows and $O(s^2U) = O(s^5) = O(m^{10/7})$ columns), with row compressions of size $n = \Theta(s \log s) = \Theta(m^{2/7} \log m) \approx N^{1/5}$. Thus, any $O(Nn^{2-\varepsilon})$ algorithm computing $M \cdot v$ would solve 3SUM instances in time $\tilde{O}(m^{2-2\varepsilon/7})$, refuting the 3SUM conjecture.

# 5 Matrix-Matrix Multiplication

In this section, we consider the problem of computing the matrix product $C$ of two $N \times N$ matrices $A, B$. We consider the following representations of the input matrices:

- **Convenient compression:** $A$ is compressed row-wise, $B$ is compressed column-wise. This representation allows us to compute any single entry $C_{i,j}$ by running an inner product algorithm on the compressed row $A_i$ and the compressed column $B_j$. The size of the input is $O(N\bar{n}_{\text{in}})$, where $\bar{n}_{\text{in}}$ is the maximum compressed size of the rows $A_i$ and columns $B_j$.
- **Strong compression:** For any matrix $M$, we define *strong compression* as a grammar compression of $M$ or $M^T$ when viewed as $n^2$-dimensional vector, whichever is shortest. When both $A, B$ are given as strong compression, the resulting representation can have a much smaller size (it can be $o(N)$), but to compute a single entry $C_{i,j}$, we first might need to obtain a representation of the row $A_i$ and the column $B_j$.

Similarly, we have several options for representing $C$:

- **Row-wise compression of $C$.** This compression is particularly useful if we aim to compute repeated matrix products $A_1(A_2(\cdots(A_kB)))$. The output size is $O(N\bar{n}_{\text{out}})$, where $\bar{n}_{\text{out}}$ is the maximum compressed size over all rows of $C$.
- **Column-wise compression of $C$.** This compression is particularly useful if we aim to compute repeated matrix products $(((AB_1)B_2)\cdots)B_k$. The output size is $O(N\bar{n}_{\text{out}})$, where $\bar{n}_{\text{out}}$ is the maximum compressed size over all columns of $C$.

- **Strong compression of** $C$**.** This compression has the smallest output size, which can be even $o(N)$.

We show the following result:

**Theorem 5.1.** *For infinitely many $N$, there are $N \times N$ matrices $A, B$ with*

1. *convenient compression of size $O(N \log N)$ (already under RLE), and*

2. *strong compression of size $O(\log^2 N)$, such that*

3. *the matrix product $C = AB$ has size $\Omega(N^2/\log^2 N)$ in any grammar-compression (row-wise, column-wise, or strong).*

As a consequence, there can be no $o(N^2/\log^2 N)$ algorithm for matrix-matrix multiplication (for any of our discussed representations), since already writing the output requires time $\Omega(N^2/\log^2 N)$.

The rough proof strategy is to construct an instance $C = AB$ such that $C$ and $C^T$, when viewed as $N^2$-dimensional vectors, contain all substrings of length $2 \log_2 n$. By the following standard lemma, such a string has no grammar compression of size $o(N^2/\log N)$.

**Lemma 5.2** (see, e.g., [20, Lemma 3]). *Let $\ell \in \mathbb{N}$. If a string $x$ is generated by a grammar of size $n$, then $x$ contains at most $n\ell$ distinct substrings of length $\ell$.*

*Proof of Theorem 5.1.* Let $\ell \in \mathbb{N}$. We first define the matrices $A', B'$ where $A'$ is a $(2^\ell \times 2\ell)$ matrix with rows indexed by strings $x \in \{0,1\}^\ell$ in lexicographic order, and $B'$ is a $(2\ell \times 2^\ell(2\ell))$ matrix with columns indexed by $(y, k) \in \{0,1\}^\ell \times \{1, \ldots, 2\ell\}$ in lexicographic order. For arbitrary $z \in \{0,1\}^\ell$, let $\mathrm{diag}(z)$ denote the $\ell \times \ell$ diagonal matrix with $z$ on the diagonal. We define

$$A'_x := (x \mid 1^\ell), \qquad B'_{(y,1),\ldots,(y,2\ell)} := \left( \begin{array}{c|c} \mathrm{diag}(1^\ell) & 0 \\ \hline 0 & \mathrm{diag}(y) \end{array} \right).$$

Let $C' = A'B'$ be the $(2^\ell \times 2^\ell(2\ell))$ product matrix of $A'$ and $B'$, with rows and columns indexed by $\{0,1\}^\ell$ and $\{0,1\}^\ell \times \{1,\ldots,2\ell\}$, respectively. Observe that by definition, $(C_{x,(y,1)}, \ldots, C_{x,(y,2\ell)}) = (x \mid y)$ for any $x, y \in \{0,1\}^\ell$. In particular, when we view $C'$ as a $2^{2\ell}(2\ell)$-length string, it contains all strings in $\{0,1\}^{2\ell}$ as substrings, thus by Lemma 5.2, any row-wise compression is of size at least $2^{2\ell}/(2\ell)$.

It is straightforward to make these matrices quadratic with dimension $N = \Theta(\ell 2^\ell)$ (by introducing all-0 columns) and to ensure that also column-wise compression has size $\Omega(2^{2\ell}/\ell) = \Omega(N^2/\log^2 N)$ (using transposed constructions to $A'$ and $B'$). Finally, we can compress each row of $A'$ and column of $B'$ trivially to length $O(\ell) = O(\log N)$ (already using RLE). In the appendix, we also argue how to grammar-compress the concatenation of the columns of $A'$ and the rows of $B'$ to size $O(\ell^2) = O(\log^2 N)$, which concludes the desired bound on the strong compression. $\square$

## Broader Impact

The broader impact of our work is to inform algorithm design for compressed linear algebra, which can lead to faster algorithms for a variety of tasks on large data sets. The ethical consequences depend on the specific application. We do not see any inherently new concerns raised by our results, beyond those that follow generally from faster algorithms and an increased ability to process data.

## Acknowledgments and Disclosure of Funding

We thank the anonymous reviewers for their helpful comments and suggestions.

*Arturs Backurs*: Supported by an NSF Grant CCF-2006806.

*Karl Bringmann:* This work is part of the project TIPEA that has received funding from the European Research Council (ERC) under the European Unions Horizon 2020 research and innovation programme (grant agreement No. 850979).

## Footnotes

[1]We use the notation $\tilde{O}(n) = n \cdot N^{o(1)}$ for near-linear time, hiding small terms such as log factors.

[2]The more standard notation is $n^{2-o(1)}$ which indicates an $\Omega(n^{1.9999})$ lower bound, no matter how close to 2 we go. That is, only mildly subquadratic algorithms are possible, e.g. by shaving log factors.

[3] We remark that *some* complexity assumption is necessary for proving the kind of results we are interested, since unconditionally proving even very weak lower bounds on the time complexity such as $\Omega(n^{1+\varepsilon})$ and even for NP-Hard problems like SAT (not to mention inner product) is far beyond current techniques [12].

[4] Strictly speaking, such a conditional lower bound of $\Omega(n^{10})$ for *highly compressible* inputs can already be proven by combining a known #P-hardness reduction from SubsetSum [60] with a fine-grained hardness of SubsetSum under the Exponential Time Hypothesis (see, e.g. [47]). However, such an approach yields only a weak lower bound in terms of the uncompressed size $N$, namely a bound of $\Omega(N^{\epsilon})$ for some non-explicit, possibly tiny $\epsilon$. Our lower bounds always give an explicit, reasonably large value for $\epsilon$.

[5]For example, instead of $a + b = c$ or $a + b + c = 0$ we may be given a target $t$ and ask for $a + b + c = t$.

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
