[Supplementary Material]

# A  Further Preliminaries

For a sequence of vectors $v_1, \ldots, v_\ell$, we let $v_1 \, v_2 \, \ldots \, v_\ell = v_1 \circ v_2 \circ \cdots \circ v_\ell = \bigcirc_{i=1}^{\ell} v_i$ denote their concatenation.

By the following observation, when proving a lower bound for a compression of size $\Theta(N^\gamma)$, the main task is to prove the upper bound $n = O(N^\gamma)$; the lower bound $n = \Omega(N^\gamma)$ can be ensured mechanically.

**Observation A.1.** *Let $0 \leqslant \gamma \leqslant 1$. Given two $N$-dimensional vectors $u, v$ of compressed size $O(N^\gamma)$, we can compute two $O(N)$-dimensional vectors $u', v'$ of compressed size $\Theta(N^\gamma)$ with the same inner product.*

*Proof.* Append $0^{N^\gamma}$ using $\Theta(N^\gamma)$ additional rules to the encodings of $u$ and $v$. $\qquad\square$

**The Strong $k$SUM Assumption**   To generalize the lower bound of Theorem 1.1 so that it works for an arbitrary relationship between compressed and uncompressed sizes, we will use an assumption about a generalized version of 3SUM.

**Definition A.2** (The $k$SUM Problem). *Given $k$ sets $A_1, \ldots, A_k$ of $m$ integers in $\{1, \ldots, U\}$, decide if there are $k$ numbers $a_1 \in A_1, \ldots, a_k \in A_k$ such that $a_1 + \cdots + a_{k-1} = a_k$.*

For all constant $k \geqslant 3$ a simple meet-in-the-middle algorithm with hashing solves $k$SUM in $O(m^{\lceil k/2 \rceil})$ time, and no faster algorithm by $m^\varepsilon$ factors, for any $\varepsilon > 0$, is known to date, unless the universe size $U$ is smaller than $O(m^{\lceil k/2 \rceil - \varepsilon})$. This is because Fast Fourier Transform gives an $O(m + kU \log U)$ time algorithm [27]. It is conjectured that substantially faster algorithms do not exist (e.g. in [4, 2]).

*The Strong $k$SUM Conjecture.* For all constant $k \geqslant 3$ it holds that: no algorithm can solve the $k$SUM problem with $U = O(m^{\lceil k/2 \rceil})$ in $O(m^{\lceil k/2 \rceil - \varepsilon})$ time, where $\varepsilon > 0$.

Observe that this assumption is about all $k \geqslant 3$ and therefore implies the Strong 3SUM conjecture as a special case. Intuitively, the reason this problem helps us give reductions where the vectors are much more compressible is that, compared to 3SUM, as $k$ grows the ratio between the time complexity $m^{k/2}$ and the input size $m$ grows.

# B  Vector Inner Product

In this section, we prove the generalization of the lower bound of Theorem 1.1 to arbitrary relationships between compressed and uncompressed sizes of the vectors.

**Theorem B.1.** *Let $0 < \varepsilon < 1/3$. Assuming the Strong $k$SUM conjecture for all constant $k$, the inner product of two $N$-dimensional vectors that are grammar-compressed to size $n = \Theta(N^\varepsilon)$ cannot be computed in $O(N^{1/3 - \delta})$ time, where $\delta > 0$.*

This result follows from the following stronger statement.

**Theorem B.2.** *Let $k \geqslant 3$. Assuming the Strong $k$SUM conjecture, the inner product of two $N$-dimensional vectors that are grammar-compressed to size $n = \Theta(N^{1/\lceil \frac{3k-4}{2} \rceil})$ cannot be computed in $O(N^{(1/3 + \gamma_k) - \delta})$ time, where $\delta > 0$ and*

$$\gamma_k := \begin{cases} \frac{2}{3(k-1)}, & \text{if } k \text{ is odd,} \\ \frac{4}{9k-12}, & \text{if } k \text{ is even.} \end{cases}$$

Observe that the above statement implies Theorem B.1: For any $0 < \varepsilon < 1/3$, we choose $k$ sufficiently large such that $1/\lceil \frac{3k-4}{2} \rceil < \varepsilon$. Then using Observation A.1, we obtain that any $O(N^{1/3 - \delta})$-time algorithm for Vector Inner Product with compressed size $n = \Theta(N^\varepsilon)$ would give an $O(N^{1/3 + \gamma_k - \delta'})$-time algorithm for Vector Inner Product with compressed size $O(N^{1/\lceil \frac{3k-4}{2} \rceil}) = O(N^\varepsilon)$, where $\delta' = \gamma_k + \delta$ – this would refute the Strong $k$SUM conjecture by Theorem B.2.

Furthermore, observe that if we set $k = 3$, we obtain a $\tilde{\Omega}(N^{2/3})$ lower bound for compressed size $n = \Theta(N^{1/3})$ under the Strong 3SUM conjecture.

In the remainder of this section, we give the proof of Theorem B.2. The central construction is captured by the following lemma.

**Lemma B.3.** *Given sets $A_1, \ldots, A_k$ of integers in $\{1, \ldots, U\}$, we define*

$$v'_{A_1 + \cdots + A_{k-1}} := \bigcirc_{\substack{(a_1, \ldots, a_{k-2}) \in A_1 \times \cdots \times A_{k-2} \\ \text{in lexicographic order}}} 0^{a_1 + \cdots + a_{k-2}} v_{A_{k-1}} 0^{(k-2)U - a_1 - \cdots - a_{k-2}},$$

$$v'_{A_k} := (v_{A_k} 0^{(k-2)U})^{m^{k-2}},$$

*where $v_{A_{k-1}}, v_{A_k} \in \{0, 1\}^U$ denote the characteristic vectors of the sets $A_{k-1}, A_k$. We have the following properties:*

1. *The inner product of the $m^{k-2}(k-1)U$-dimensional vectors $v'_{A_1 + \cdots + A_{k-1}}$ and $v'_{A_k}$ is nonzero if and only if there is a tuple $(a_1, \ldots, a_k) \in A_1 \times \cdots \times A_k$ with $a_1 + \cdots + a_{k-1} = a_k$.*

2. *We can compute compressions of $v'_{A_1 + \cdots + A_{k-1}}, v'_{A_k}$ of size $O(km \log U) = O(m \log U)$ in time $O(m \log U)$.*

*Proof.* For 1., observe that by construction, $v'_{A_1 + \cdots + A_{k_1}}$ and $v'_{A_k}$ consist of $m^{k-2}$ blocks, indexed by $(a_1, \ldots, a_{k-2}) \in A_1 \times \cdots \times A_{k-2}$ and consisting of the sequence $0^{a_1 + \cdots + a_{k-2}} v_{A_{k-1}} 0^{(k-2)U - a_1 - \cdots - a_{k-2}}$ and $v_{A_k} 0^{(k-2)U}$ of length $(k-1)U$, respectively. In particular, in block $(a_1, \ldots, a_{k-2})$ there is a common 1-entry $t$ if and only if $t = (a_1 + a_2 + \cdots + a_{k-2}) + a$ for some $a \in A_{k-1}$ and $t = a'$ for some $a' \in A_k$. Thus, there exists a common 1-entry in $v'_{A_1 + \cdots + A_{k-2}}$ and $v'_{A_k}$ if and only if there are $(a_1, \ldots, a_k) \in A_1 \times \cdots \times A_k$ with $a_1 + \cdots + a_{k-1} = a_k$.

For 2., we first recall that as shown in the proof of Theorem 1.1, we can compute a compression of the characteristic vectors $v_{A_{k-1}}$ and $v_{A_k}$ of size $O(m \log U)$ in time $O(m \log U)$. Thus, using Proposition 2.1, we can compute a compression of $v'_{A_k} = (v_{A_k} 0^{(k-2)U})^{m^{k-2}}$ of size $O(m \log U) + O(\log((k-2)U)) + O(\log m^{k-2}) = O(m \log U)$ in time $O(m \log U)$. To show the claim for $v'_{A_1 + \cdots + A_{k-1}}$, we proceed inductively and construct the strings $v'_{A_{k-1}} := v_{A_{k-1}}$ and

$$v'_{A_i + \cdots + A_{k-1}} := \bigcirc_{\substack{(a_i, \ldots, a_{k-2}) \in A_i \times \cdots \times A_{k-2} \\ \text{in lexicographic order}}} 0^{a_i + \cdots + a_{k-2}} v_{A_{k-1}} 0^{(k-1-i)U - a_i - \cdots - a_{k-2}},$$

for $i = k-2, \ldots, 1$. The central observation is that we can write $A_i = \{a_1^{(i)}, \ldots, a_m^{(i)}\}$ with $a_1^{(i)} < a_2^{(i)} < \cdots < a_m^{(i)}$ and obtain

$$v'_{A_i + \cdots + A_{k-1}} = \bigcirc_{j=1}^m 0^{a_j^{(i)}} v'_{A_{i+1} + \cdots + A_{k-1}} 0^{U - a_j^{(i)}}.$$

Thus, given an SLP $\mathcal{G}_{i+1}$ for $v'_{A_{i+1} + \cdots + A_{k-1}}$ with starting symbol $S_{i+1}$, we can give an SLP $\mathcal{G}_i$ for $v'_{A_i + \cdots + A_{k-1}}$ of size $|\mathcal{G}_{i+1}| + O(m \log U)$ as follows: For each $j = 1, \ldots, m$, we encode $0^{a_j^{(i)}}$ using $O(\log a_j^{(i)}) = O(\log U)$ additional symbols, re-use $S_{i+1}$ to generate $v'_{A_{i+1} + \cdots + A_{k-1}}$, and encode $0^{U - a_j^{(i)}}$ using $O(\log(U - a_j^{(i)})) = O(\log U)$ additional symbols. Observe that we can obtain this compression in time $O(m \log U)$.

Thus, starting from an SLP for $v'_{A_{k-1}}$, after $k-2$ steps we obtain an SLP $\mathcal{G}_1$ for $v'_{A_1 + \cdots + A_{k-1}}$ of size $O(km \log U) = O(m \log U)$. The running time of this construction is $O(km \log U) = O(m \log U)$, concluding the proof. □

Let $A_1, \ldots, A_k \subseteq \{1, \ldots, U\}$ be a Strong $k$SUM instance, i.e., $U = O(m^{\lceil k/2 \rceil})$. The reduction given in Lemma B.3 gives two vectors $v, v'$ of dimension $m^{k-2} \cdot (k-1)U$ such that their inner product allows us to decide the $k$SUM instance. Furthermore, the vectors have a compressed size of $O(m \log U)$.

We slightly adapt $v, v'$ by appending 0's to increase the dimension slightly to $N = m^{k-2} \cdot (k-1)U \log^{\lceil (3k-4)/2 \rceil} U$ (this does not change their inner product). We verify the following facts: (1) an $O(N^{1/3 + \gamma_k - \delta})$-time Vector Inner Product algorithm for some $\delta > 0$ refutes the Strong $k$SUM

conjecture and (2) $n = O(N^{1/\lceil \frac{3k-4}{2} \rceil})$. Using Observation A.1, this concludes the proof of Theorem B.2.

For (1), consider first the case that $k$ is odd. Then $U = O(m^{(k+1)/2})$ and $N = O(m^{k-2}U \mathrm{polylog} U) = O(m^{3(k-1)/2} \mathrm{polylog} m)$. Observe that

$$N^{1/3+\gamma_k-\delta} = O(m^{\frac{3(k-1)}{2} \cdot (\frac{1}{3} + \frac{2}{3(k-1)} - \delta)} \mathrm{polylog} m)$$
$$= O(m^{\frac{k-1}{2}+1-\frac{3(k-1)}{2}\delta}) = O(m^{\lceil \frac{k}{2} \rceil - \delta'}),$$

for any $0 < \delta' < 3(k-1)\delta/2$.

Similarly, for even $k$, we have $U = O(m^{k/2})$ and $N = O(m^{k-2}U\mathrm{polylog} U) = O(m^{(3k-4)/2} \mathrm{polylog} m)$. Using $1/3 + \gamma_k = 1/3 + 4/(9k-12) = k/(3k-4)$, we obtain that

$$N^{1/3+\gamma_k-\delta} = O(m^{\frac{3k-4}{2} \cdot (\frac{k}{3k-4} - \delta)} \mathrm{polylog} m) = O(m^{\frac{k}{2} - \delta'}),$$

for any $0 < \delta' < (3k-4)\delta/2$. Thus, in both cases, an $O(N^{1/3+\gamma_k-\delta})$-time Vector Inner Product algorithm refutes the Strong $k$SUM conjecture by solving the given $k$SUM instance in time $O(m^{\lceil k/2 \rceil - \delta'})$ with $\delta' > 0$.

Finally, for (2), note that $N = O(m^{k-2}U \log^{\lceil (3k-4)/2 \rceil} U) = O(m^{\lceil (3k-4)/2 \rceil} \log^{\lceil (3k-4)/2 \rceil} m)$. Thus $n = O(m \log m) = O(N^{1/\lceil (3k-4)/2 \rceil})$, as desired.

## C   Matrix-Vector Product

In this section we provide the full proof of Theorem 1.2. We first prove a self-reduction for 3SUM as a central tool (using standard techniques), and then proceed to give the final reduction.

### C.1   Proof of the Self-Reduction

Let us restate Lemma 4.1.

**Lemma C.1** (Self-Reduction for 3SUM). *Let $1 \leqslant s = s(m) \leqslant m$ and $\varepsilon > 0$ be arbitrary. If there is an algorithm that, given a target $t$ and $L = O((m/s)^2)$ sets $A_\ell, B_\ell, C_\ell$ of $s$ integers in $\{1, \ldots, O(s^3 \log^2 s)\}$, determines for all $1 \leqslant \ell \leqslant L$ whether there are $a \in A_\ell, b \in B_\ell, c \in C_\ell$ with $a + b + c = t$ in total time $O(m^{2-\epsilon})$, then the 3SUM conjecture is false.*

In the remainder of this section, we give the proof.

Let $A, B, C$ be sets of $m$ integers in $\{1, \ldots, U\}$. We use a couple of results from earlier work that are stated for the following 3SUM formulation: given three sets $A', B', C'$ of $m$ integers in $\{-U, \ldots, U\}$ with $U = O(m^3 \log^2 m)$, we are asked to determine whether there are $a \in A', b \in B', c \in C'$ such that $a + b + c = 0$. We first reduce our formulation to this formulation by setting $A' := A, B' := B$, and $C' := -C = \{-c \mid c \in C\}$. We can now use the following known self-reduction for 3SUM.

**Lemma C.2** (Reformulated from [62, Theorem 13]). *Let $s := s(m)$ with $1 \leqslant s \leqslant m$. Given three sets $A', B', C'$ of $m$ integers in $\{-U, \ldots, U\}$, we can compute, in time $O(m^2/s)$, a list of $L = O((m/s)^2)$ 3SUM instances, i.e., sets $A'_\ell, B'_\ell, C'_\ell$ with $1 \leqslant \ell \leqslant L$, such that there is an $a \in A', b \in B', c \in C'$ with $a + b + c = 0$ if and only if there is an instance $1 \leqslant \ell \leqslant L$ and a triple $a \in A'_\ell, b \in B'_\ell, c \in C'_\ell$ with $a + b + c = 0$. Furthermore, each $A'_\ell, B'_\ell, C'_\ell$ is a subset of $s$ integers of $A', B', C'$, respectively.*

*Proof sketch.* We give the high-level arguments (for details, see the proof of Theorem 13 in [62]). For a set $S$, let $\min S$ and $\max S$ denote the smallest and largest element in $S$, respectively. We sort $A', B', C'$ and split each array into $\lceil m/s \rceil$ consecutive parts $A'_1, \ldots, A'_{\lceil m/s \rceil}, B'_1, \ldots, B'_{\lceil m/s \rceil}, C'_1, \ldots, C'_{\lceil m/s \rceil}$, each of at most $s$ elements, such that $\max A'_i < \min A'_{i+1}, \max B'_i < \min B'_{i+1}$ and $\max C'_i < \min C'_{i+1}$ for all $i$. Instead of searching for a 3SUM triple $a \in A'_i, b \in B'_j, c \in C'_k$ for each $1 \leqslant i, j, k \leqslant \lceil m/s \rceil$ (i.e., $\Theta((m/s)^3)$ subproblems with $s$ elements each), one observes that most subproblems can be trivially solved: We say that a subproblem $(i, j, k)$ is trivial, if $\min A_i + \min B_j + \min C_k > 0$ or $\max A_i + \max B_j + \max C_k < 0$;

these subproblems cannot contain a solution. The key insight is that there are at most $O((m/s)^2)$ non-trivial subproblems (which follows since the *domination* partial ordering on $\{1, \ldots, u\}^3$ has at most $O(u^2)$ incomparable elements); these can be determined in time $O((m/s)^2)$. Thus, it suffices to list all $O((m/s)^2)$ non-trivial subproblems with $s$ integers in each set in time $O(m^2/s)$. $\qquad\square$

The resulting instances $A'_\ell, B'_\ell, C'_\ell$ consist of integers in $\{-U, \ldots, U\}$ with large universe size $U = O(m^3 \log^2 m)$. We reduce the universe size to $O(s^3 \log^2 s)$ using a folklore technique (a slightly stronger result with $U = O(s^3)$ can be achieved using the techniques of [15]). To prepare notation, for any set $S$, we let $S \bmod p := \{s \bmod p \mid s \in S\}$.

**Lemma C.3** (Adaptation of [5, Lemma B.1])**.** *There is some $\alpha$ such that $U' := \alpha s^3 \log s \log U$ satisfies the following property: Let $A, B, C$ be sets of $s$ integers in $\{-U, \ldots, U\}$ such that no $a \in A, b \in B, c \in C$ satisfies $a + b + c = 0$. Let $p$ be a prime chosen uniformly at random from $\{2, \ldots, U'\}$. Then the probability that there are $a_p \in A \bmod p, b_p \in B \bmod p, c_p \in C \bmod p$ with $a_p + b_p + c_p \equiv 0 \pmod{p}$ is at most $1/2$.*

*Proof.* Let $a \in A, b \in B, c \in C$ be arbitrary. Since $a + b + c \neq 0$, note that $(a \bmod p) + (b \bmod p) + (c \bmod p) \equiv 0 \pmod{p}$ if and only if $p$ divides $a + b + c$. Since $a + b + c \in \{-3U, \ldots, 3U\}$, $a + b + c$ has at most $\log_2(3U)$ prime factors. Let $P$ denote the number of prime numbers in $\{2, \ldots, U'\}$; by the prime number theorem we can choose $\alpha$ large enough such that $P \geqslant 2s^3 \log_2(3U)$. Thus, the probability that $p$ was chosen among these at most $\log_2(3U)$ prime factors is at most $\log_2(3U)/P \leqslant 1/(2s^3)$. Thus, by a union bound over all $s^3$ triples $a \in A, b \in B, c \in C$, the probability that there are $a_p \in A \bmod p, b_p \in B \bmod p, c_p \in C \bmod p$ with $a + b + c \equiv 0 \pmod{p}$ is at most $1/2$. $\qquad\square$

Note that if $A, B, C$ contain a triple $a, b, c$ with $a + b + c = 0$, then also $A \bmod p, B \bmod p, C \bmod p$ contain a triple $a_p, b_p, c_p$ with $a_p + b_p + c_p \equiv 0 \pmod{p}$ for any $p$.

We can finally prove Lemma C.1: Assume that there is an algorithm $\mathcal{A}$ that given a target $t$ and $L = O((m/s)^2)$ instances $A_\ell, B_\ell, C_\ell, 1 \leqslant \ell \leqslant L$ of $s$ integers in $\{1, \ldots, U'\}$, determines for all $1 \leqslant \ell \leqslant L$ whether there are $a \in A_\ell, b \in B_\ell, c \in C_\ell$ with $a + b + c = t$ in total time $O(m^{2-\varepsilon})$ with $\varepsilon > 0$. Observe that since $\mathcal{A}$ runs in time $O(m^{2-\varepsilon})$, we must have $s = \Omega(m^\varepsilon)$, since otherwise already the size of the input to $\mathcal{A}$ of $\Theta(m^2/s)$ would be $\omega(m^{2-\varepsilon})$. Thus, we have $U' = O(s^3 \log^2 s)$.

For $r = 1, \ldots, \gamma \log m$ many repetitions, we do the following: We choose a random prime $p_r \in [2, U']$ and obtain $\ell$ instances in $\{0, \ldots, p_r - 1\} \subseteq \{0, \ldots, U'\}$ by taking the sets modulo $p_r$, i.e., $A_\ell^{(r)} := A'_\ell \bmod p_r, B_\ell^{(r)} := B'_\ell \bmod p_r$, and $C_\ell^{(r)} = C'_\ell \bmod p_r$. Observe that we may determine whether there is some $a \in A_\ell^{(r)}, b \in B_\ell^{(r)}, c \in C_\ell^{(r)}$ with $a + b + c \equiv 0 \pmod{p_r}$ by testing for each $t \in \{0, p_r, 2p_r\}$, whether there $a \in A_\ell^{(r)}, b \in B_\ell^{(r)}, c \in C_\ell^{(r)}$ with $a + b + c = t$. Thus, to do this, and additionally ensure that each integer is in $\{1, \ldots, U'\}$, we add 1 to each integer in $A_\ell^{(r)}, B_\ell^{(r)}, C_\ell^{(r)}$ and for each $\lambda \in \{0, 1, 2\}$, call $\mathcal{A}$ on the sets $A_\ell^{(r)}, B_\ell^{(r)}, C_\ell^{(r)}, 1 \leqslant \ell \leqslant L$ with common target $t_\lambda := 3 + \lambda p_r$.

Observe that after these $3\gamma \log m$ calls to $\mathcal{A}$, we know for each $1 \leqslant \ell \leqslant L$ and $1 \leqslant r \leqslant \gamma \log m$ whether there are $a \in A'_\ell, b \in B'_\ell, c \in C'_\ell$ with $a + b + c \equiv 0 \pmod{p_r}$. We declare our original 3SUM instance $A, B, C$ to be a YES instance if and only if there is some $\ell$ such that for all $r$ we have found a witness $a \in A'_\ell, b \in B'_\ell, c \in C'_\ell$ with $a + b + c \equiv 0 \pmod{p_r}$. Note that if $A, B, C$ is a YES instance, we always return YES by Lemma C.2. Otherwise, if $A, B, C$ is a NO instance, consider a fixed $\ell$. By Lemmas C.2 and C.3, the probability that for all $r$, we find $a \in A'_\ell, b \in B'_\ell, c \in C'_\ell$ with $a + b + c \equiv 0 \pmod{p_r}$ is bounded by $2^{-\gamma \log m} = m^{-\gamma}$. Thus, by a union bound over all $\ell$, the probability that we incorrectly return YES in this case is at most $Lm^{-\gamma} = O((m/s)^2 m^{-\gamma}) = O(m^{2-\gamma})$. We can make this error probability polynomially small by choosing $\gamma > 2$.

Observe that the running time of the above process is $O(\log m)$ times the running time of $\mathcal{A}$ (note that the running time used for Lemma C.2 is linear in its output size, which is the input size of $\mathcal{A}$ and thus dominated by the running time of $\mathcal{A}$). Thus, we can solve any 3SUM instance in time $O(m^{2-\varepsilon} \log m)$, which would refute the 3SUM conjecture. This concludes the proof of Lemma C.1.

## C.2 Main Reduction for Matrix-Vector Multiplication

We now turn to the proof of Theorem 1.2.

*Proof.* Let $s$ be a parameter to be chosen later. By Lemma 4.1, it suffices to solve $L = O((m/s)^2)$ 3SUM instances $A_\ell, B_\ell, C_\ell$ consisting of $s$ integers in $\{1, \ldots, U\}, U = O(s^3 \log^2 s)$ with common target $1 \leqslant t \leqslant 3U$ in time $O(m^{2-\epsilon})$ for some $\epsilon > 0$ to contradict the 3SUM conjecture.

We construct an $(L \times 3s^2 U)$ matrix $M$ and $v \in \{0,1\}^{3s^2 U}$ as follows. Intuitively, each row $M_\ell$ and the vector $v$ are partitioned into $s^2$ blocks of size $3U$. Each block is indexed by $(i,j)$ with $i, j \in \{1, \ldots, s\}$ in lexicographic order and the block of $M_\ell$ corresponding to $(i,j)$ encodes the characteristic vector of the set $a_i + b_j + C_\ell = \{a_i + b_j + c \mid c \in C_\ell\} \subseteq \{1, \ldots, 3U\}$, where $a_i$ is the $i$-th integer in $A_\ell$ and $b_j$ is the $j$-th integer in $B_\ell$. Correspondingly, every block $(i,j)$ in $v$ encodes the characteristic vector of the singleton set $\{t\} \subseteq \{1, \ldots, 3U\}$. Thus, there is a position in block $(i,j)$ in which both $M_\ell$ and $v$ have a 1 if and only if there is a $c \in C_\ell$ such that $a_i + b_j + c = t$.

Formally, for any $1 \leqslant \ell \leqslant L$, we write $A_\ell = \{a_1^\ell, \ldots, a_s^\ell\}, B_\ell = \{b_1^\ell, \ldots, b_s^\ell\}$ and define

$$M_\ell := \underbrace{0^{a_1+b_1} v_{C_\ell} 0^{3U-a_1-a_2}}_{v_{a_1^\ell + b_1^\ell + C_\ell}} \quad \cdots \quad \underbrace{0^{a_i+b_j} v_{C_\ell} 0^{3U-a_i-b_j}}_{v_{a_i^\ell + b_j^\ell + C_\ell}} \quad \cdots \quad \underbrace{0^{a_s+b_s} v_{C_\ell} 0^{3U-a_s-b_s}}_{v_{a_s^\ell + b_s^\ell + C_\ell}},$$

$$v := 0^{t-1} 1 0^{3U-t} \quad \cdots \quad 0^{t-1} 1 0^{3U-t} \quad \cdots \quad 0^{t-1} 1 0^{3U-t},$$

where $v_{C_\ell} \in \{0,1\}^U$ denotes the characteristic vector of $C_\ell$. By this structure, it is clear that $M_\ell v \geqslant 1$ if and only if there are $a \in A_\ell, b \in B_\ell, c \in C_\ell$ with $a + b + c = t$.

We will show that each row $M_\ell$ can be compressed to size $\Theta(s \log s)$ (as opposed to its RLE of length $\Theta(s^3 \log s)$). We thus will set $N = \lceil 3s^2 U \log^3 s \rceil = \Theta(s^5 \log^5 s)$, and append $0^{N-3s^2 U}$ to each row $M_\ell$ and $v$, so that we obtain an $L \times N$ matrix $M'$ and $N$-dimensional vector $v'$ whose product $M'v'$ can be used to solve all instances $A_\ell, B_\ell, C_\ell$ in linear time. Observe that each row has a compression of size $\Theta(N^{1/5}) = \Theta(s \log s)$, as desired. Since $L = O((m/s)^2)$ and $N \geqslant s^5$, we can set $s = \Theta(m^{2/7})$ such that $L \leqslant N$ (we can indeed make $L = N$ by introducing zero rows, if necessary). Thus, an $O(Nn^{2-\epsilon})$-time algorithm for multiplying $M'$ and $v'$ would solve all $L$ 3SUM instances in time

$$O(Nn^{2-\epsilon}) = O((m/s)^2 (s \log s)^{2-\epsilon}) = O((m^2/s^\epsilon) \text{polylog} s) = O(m^{2-\frac{2}{7}\epsilon} \text{polylog} m),$$

which would refute the 3SUM conjecture.

Analogous to the proof of Theorems 1.1 and B.2, we can compute a compression of size $\Theta(s \log s)$ in time $O(s \log s)$. Indeed, for each $M_\ell$, this already follows from Lemma B.3 when setting $A_1 := A_\ell, A_2 := B_\ell, A_3 := C_\ell$, which shows how to compress the string $v'_{A_1 + A_2 + A_3} = M_\ell$ to size $O(s \log U) = O(s \log s)$ in time $O(s \log U) = O(s \log s)$. For $v$, we simply apply Proposition 2.1 to the straightforward compression of $0^{t-1} 1 0^{3U-t}$ to size $O(\log U)$, which leads to a compression of $v$ of size $O(\log U + \log s) = O(\log s)$. Using Observation A.1, we can make all encodings have size $\Theta(s \log s)$, which concludes the proof. $\square$

# D Matrix-Matrix Product

In this section, we give the full proof of Theorem 5.1.

*Proof of Theorem 5.1.* Let $\ell \in \mathbb{N}$. We first define the matrices $A', B'$ where $A'$ is a $(2^\ell \times 2\ell)$ matrix with rows indexed by strings $x \in \{0,1\}^\ell$ in lexicographic order, and $B'$ is a $(2\ell \times 2^\ell (2\ell))$ matrix with columns indexed by $(y, k) \in \{0,1\}^\ell \times \{1, \ldots, 2\ell\}$ in lexicographic order. For arbitrary $z \in \{0,1\}^\ell$, let $\text{diag}(z)$ denote the $\ell \times \ell$ diagonal matrix with $z$ on the diagonal. We define

$$A'_x := (x \mid 1^\ell), \qquad B'_{(y,1),\ldots,(y,2\ell)} := \left( \begin{array}{c|c} \text{diag}(1^\ell) & 0 \\ \hline 0 & \text{diag}(y) \end{array} \right).$$

Let $C' = A'B'$ be the $(2^\ell \times 2^\ell(2\ell))$ product matrix of $A'$ and $B'$, with rows and columns indexed by $\{0,1\}^\ell$ and $\{0,1\}^\ell \times \{1, \ldots, 2\ell\}$, respectively. Observe that by definition, $(C_{x,(y,1)}, \ldots, C_{x,(y,2\ell)}) = (x \mid y)$ for any $x, y \in \{0,1\}^\ell$. In particular, when we view $C'$ as a $2^{2\ell}(2\ell)$-length string, it contains all strings in $\{0,1\}^{2\ell}$ as substrings, thus by Lemma 5.2, any row-wise compression is of size at least $2^{2\ell}/(2\ell)$.

To also ensure column-wise incompressibility, we slightly extend the construction by analogous transposed constructions: We let $N := 2^\ell(2\ell + 1)$ and define the final $(N \times N)$ matrices $A, B$ as follows:

$$A := \left( \begin{array}{c|c|c} A' & 0 & 0 \\ \hline 0 & B'^T & 0 \end{array} \right), \qquad\qquad B := \left( \begin{array}{c|c} B' & 0 \\ \hline 0 & A'^T \\ \hline 0 & 0 \end{array} \right).$$

Since $C := AB = \left( \begin{array}{c|c} A'B' & 0 \\ \hline 0 & (A'B')^T \end{array} \right)$ contains all length-$(2\ell)$ strings as substrings of the rows (in the $A'B'$ part) and as substrings of the columns (in the $(A'B')^T$ part), any strong compression of $C$ is of size at least $2^{2\ell}/(2\ell) = \Omega(N/\log^2 N)$, proving the third part of the claim.

For the first two parts, it remains to show that $A$ and $B$ can be well compressed: For the convenient compression, we observe that any row in $A$ is either of the form $(x1^\ell \mid 0^{2\ell} \mid 0^{N-4\ell})$, which has a RLE of length at most $|x1^\ell| + O(\log N) = O(\log N)$, or it is of the form $(0^{2\ell} \mid 0^{i-1}\alpha 0^{2\ell - i} \mid 0^{N-4\ell})$ for some $\alpha \in \{0,1\}, i \in \{1, \ldots, 2\ell\}$, which also has a RLE of length at most $O(\log N)$. Thus, each of the $N$ rows of $A$ can be compressed to size $O(\log N)$, as desired. By a symmetric statement, also each column of $B$ has a RLE of size $O(\log N)$.

Finally, for the strong compression, we show that we compress $A^T$ when viewed as a string, i.e., we compress the concatenation of the columns of $A$. The main insight is the following: Imagine a binary $\ell$-bit counter. Using grammar compression, we can compress the sequence of values of any fixed bit while the counter counts from $0$ to $2^\ell - 1$ in size $O(\ell)$. Formally, let $G_0, G_1$ be grammar compressions of strings $s_0, s_1$. For any $1 \leqslant i \leqslant \ell$, we can encode $(s_0^{2^{\ell-i}} s_1^{2^{\ell-i}})^{2^{i-1}}$ using only $O(\ell)$ additional non-terminals in the canonical way. Specifically, using $O(\ell - i)$ new symbols, we may encode $s_0^{2^{\ell-i}} s_1^{2^{\ell-i}}$; let $\tilde{S}$ denote the corresponding non-terminal. We then encode $\tilde{S}^{2^{i-1}}$ using $O(i)$ additional new symbols. In total, we only need $O((\ell-i)+i) = O(\ell)$ additional symbols, as desired.

We apply the above idea to encode the concatenation all columns of $A$ as follows: Consider column $i$.

- For $1 \leqslant i \leqslant \ell$, then by the chosen lexicographic order of the row indices $x \in \{0,1\}^\ell$ of $A'$, note that the $i$-th column of $A$ is of the form $(0^{2^{\ell-i}} 1^{2^{\ell-i}})^{2^{i-1}} \mid 0^{N-2^\ell}$. Using the above analysis, we can compress it to size $O(\ell) + O(\log N) = O(\log N)$.

- If $\ell + 1 \leqslant i \leqslant 2\ell$, the $i$-th column is of the form $1^{2^\ell} \mid 0^{N-2^\ell}$, which we can compress to size $O(\log \ell + \log N) = O(\log N)$.

- If $2\ell + 1 \leqslant i \leqslant 3\ell$, write $i = 2\ell + i'$ and observe that the $i$-th column of $A$ is of the form $0^{2^\ell} \mid (0^{i'-1}10^{\ell-i'})^{2^\ell}$. Using $O(\ell)$ non-terminals to encode $0^{i'-1}10^{\ell-i'}$, it is immediate that we can compress the complete column using $O(\ell)$ additional non-terminals, i.e., yielding a total of $O(\ell) = O(\log N)$.

- If $3\ell + 1 \leqslant i \leqslant 4\ell$, write $i = 3\ell + i'$ and observe that by the chosen lexicographic order of the column indices $(y,k) \in \{0,1\}^\ell \times \{1, \ldots, 2\ell\}$ of $B'$, the $i$-th column of $A$ is of the form $0^{2^\ell} \mid (s_0^{2^{\ell-i'}} s_1^{2^{\ell-i'}})^{2^{i'-1}}$ where $s_\alpha := 0^{i'-1}\alpha 1^{\ell-i'}$. We can give trivial grammars of size $O(\ell)$ for $s_0, s_1$. Then, by the above analysis, we only need $O(\ell)$ additional non-terminals for the counter-like part. In total, we only need $O(\ell) = O(\log N)$ non-terminals to encode the $i$-th column.

- Finally, observe that the remaining columns $i = 4\ell + 1, \ldots, N$ consist of $(N - 4\ell)N$ zeroes, which we can encode together using only $O(\log N)$ non-terminals.

In summary, we can encode the first $4\ell$ columns using $O(\log N)$ non-terminals each, and only $O(\log N)$ non-terminals for the remaining columns, so we can fully compress the concatenation of $A$'s columns to size $O(\log^2 N)$, as claimed. $\qquad\square$