[Reviews · NeurIPS 2020]

Review 1

Summary and Contributions: The submission studies the complexity of computing vector-vector, matrix-vector, and matrix-matrix inner products when the input is grammar-compressed. For the first two problems, the authors obtain running time lower bounds that have quadratic (rather than linear) dependence on the compressed size, assuming the 3SUM hypothesis. For matrix-matrix multiplication, they show that even if A and B have small grammar-compression, their product may not (hence writing it would require at least linear time).

Strengths: • The problem introduced in this paper is very natural, and the connection to fine grained complexity and in particular 3SUM hypothesis is somewhat surprising. • The writing is fantastic.

Weaknesses: • The results only hold for exact computation. The authors address this by pointing out that (i) in ML applications even a small error can propagate to a big error downstream; and (ii) no multiplicative approximation is possible since they show hardness for 0-vs-1 inner product. I like this explanation. • The fine-grained complexity lower bounds are polynomial (specifically, quadratic) but it’s unless I missed something it’s not obvious that these problems are even in P. • The authors mention that a “slightly weaker” result than the vector-vector result (which IMHO is the main result) can be extracted from previous work. Please elaborate on that. **** Post feedback/discussion **** Now that I have a better understanding of context and previous results, I'm not as impressed. The high level takeaway message for someone trying to implement fast compressed vector-vector products probably doesn't change much between O(N^{1/3}) and O(N^{2/3}). And neither of them is tight anyway.

Correctness: Yes

Clarity: Fantastic

Relation to Prior Work: Yes, except aforementioned comment about "slightly weaker" result from previous work.

Reproducibility: Yes

Additional Feedback: • As much as I love FGC, I think that “efficient = polynomial time” is far from obsolete. If anything, as both data and compute power grow, the difference between polynomial and exponential becomes more obvious (think 10^3 vs 2^10, and 50^3 vs 2^50). • Matrix multiplication theorem: C cannot be **grammar**-compressed. Of course, it can be compressed as “product of two compressed matrices”. • It’s inaccurate that “all results in fine-grained complexity” are against both deterministic and randomized algorithms. It’s true as long as the fine-grained hardness assumption is assumed against randomized algorithms, and sometimes that’s obviously false, e.g. in the case of [Abboud and Backurs, ITCS 2017]. • Footnote 3: “the kinds of results we are interested” -> “the kind of results we are interested in”. • Exploiting the structure of non-worst-case is not “impossible”. Maybe “challenging” or “seems beyond current reach”. • It seems that there is no Broader Impact section. It seems reasonable given the nature of submission, but according to CFP you had to explicitly write that.


Review 2

Summary and Contributions: The paper considers the possibility of running algorithms directly on the compressed data to obtain significant time savings. In particular, the paper considers the compression with restricted form of grammar compressed strings that capture modern compression tools like Lempel-Ziv. Let N be the input size and T(N) = n be the compressed size. The goal would be to create algorithms with running time that depend on n in the same way standard algorithms depend on N. In this paper the authors consider dot product, matrix vector product and matrix matrix product and show conditional lower bounds by reduction from problems assumed to be hard (3SUM, K-SUM) For matrix-matrix product, the authors show that even when the input matrices can be greatly compressed the output (in compresses form) still requires essentially N^2 bits, which means that any algorithm working on compressed data would need at least this time. For dot product of two vectors, the authors show several results for different assumptions. The gist of it is that any algorithm would need N^p for some p < 1 time to compute the inner product, where the highest p shown is 2/3, and that in terms of compressed size n, for some compression rates even n^10 time is not enough. Finally for matrix vector, the authors show that N n^2 time is required, for a compression of matrix row and the query vector to n = N^{1/5} ). This translates to N^{7/5} conditional lower bound. In total the results show that we cannot get algorithms in the general case that work directly on top of highly compressed (like zip) data for these basic problem and get running time that essentially match the running algorithms designed to work with uncompressed data in terms of input data size.

Strengths: The strong point is a simple argument that we cannot create general purpose algorithms that works efficiently in terms of compressed data size. The proofs contained in the paper are quite elegant.

Weaknesses: The gist of the result for dot product follows from earlier work [2]. The bounds still leave a lot of room for improvement. An N^{2/3} algorithm for dot product would be pretty great, same for matrix vector (N^{7/5} and matrix multiplication (N^2) as well. Proofs shown are fairly straight forward and the results are not that surprising especially considering the work in [2]. **** Post feedback/discussion **** The message of the result follows from earlier work, the new bounds are still not tight and the improvements and added results does not change that message. While the improvements are of technical interest i do not believe that warrants publication at Neurips, hence my lowered score. It also seems that the existing np hardness (or #P hardness) result could be discussed more. Combining that result with assuming SETH and padding it seems that n^{1+\Omega(1)} and N^{\Omega(1)} should follow for polynomial compression as well.

Correctness: The proofs in the submission are fine. I only checked proofs in the submission and not in the supplemental material but i have no doubt to that the claims proved in the supplemental material should not be true.

Clarity: In general i would say yes. However, the paper uses more 4 pages for introduction instead of using some of the pages to show more of their work.

Relation to Prior Work: Yes

Reproducibility: Yes

Additional Feedback:


Review 3

Summary and Contributions: The paper shows that certain approaches to speeding up linear algebra operations, based on grammar compression, cannot be made computationally efficient (under a common hardness assumption)

Strengths: - The paper is scholarly, well-written, and technically interesting - The problem it studies is of some interest to the NeurIPS community

Weaknesses: The paper is "doubly theoretical" in the sense that: 1) It is not shown empirically that vectors and matrices encountered in ML are susceptible to grammar compression that is more effective than other (simpler) compression methods. For example, reference [3] in the rebuttal shows a simple method that has higher compression than grammar-based gzip. 2) The hardness results are for certain structured vectors only, and it is unclear if they would apply in realistic settings.

Correctness: Yes, I have no reason to doubt any of the theoretical claims. There is no empirical investigation. It would be particularly interesting to do an empirical investigation of compression of matrices with model parameters, since this often dominates the space. In this setting, compression is often achieved by some kind of lossy quantization, followed by lossless compression. See e.g.: Han S, Mao H, Dally WJ. Deep compression: compressing deep neural networks with pruning, trained quantization and Huffman coding. ICLR 2016 Tejalal Choudhary, Vipul Mishra, Anurag Goswami & Jagannathan Sarangapani. A comprehensive survey on model compression and acceleration Artificial Intelligence Review (2020)

Clarity: Yes

Relation to Prior Work: There is room for improvement, but based on the rebuttal I think the authors can make a clear discussion.

Reproducibility: Yes

Additional Feedback: Your paper would be more interesting if you considered approximate computation in more detail beyond 0 vs 1 inner product ruling out multiplicative approximation. Machine learning methods are usually robust to small, additive error. Another comment: the ordering of dimensions is arbitrary, and grammar compression (which is aimed at sequences) does not seem like a natural, or robust, choice.


Review 4

Summary and Contributions: This theoretical paper shows an impossibility result on the computational time of grammar-compressed vector and matrix products. In particular they show a worst-case upper bound on the time it requires to compute products of vector and matrices that are grammar-compressed. POS-DISCUSSION We had a rich discussion: I still believe that this paper has the potential for high impact in a reduced community. While the paper is is very well explained, more success stories of compressed ML will help bring the message across

Strengths: This is a strong paper which has the influence of deciding on future research programmes. Operating on compressed data is a tempting direction in a field which is becoming more and more data-hungry. The success in pattern matching on processing directly on compressed data showed that it is possible to obtain more efficient algorithm that operate directly on the compressed data, without the need of decompressing first. Many times, those algorithms run faster because the total length of the input is smaller. At the same time grammar-based compressor are a powerful and vast family of algorithms (including Byte-Pair Encoding for a recent example in ML), with strong optimal approximation theorems. In this sense, proving a limitation of those models will greatly influence future research programmes.

Weaknesses: This paper will not be of appeal to the empirical crowd attending NeurIPS. In particular, despite the huge amount of data modern deep learning are consuming, it has not become mainstream so far to operate directly on the compressed version. Also, worst-case analysis have hardly stopped modern machine learning to pursue directions that work in practice. However, this paper will discourage certain directions to the profit of others (eg other compression techniques).

Correctness: The proof is done by constructing one example that fulfills the conditions of the Theorem.

Clarity: The author do a great job providing the intutions in the introduction and insiting on the possible consequences of their theorems, as well as pointing out the limitations. I applaud the effort the authors took in being pedagogical, although sometimes it is a bit excessive (eg: lines 108-113). Sections 3-5 are inherently dry, but there are clearly separated and introduced. They also contain only proofs for simpler cases, pushing the proff of the general case to the appendix.

Relation to Prior Work: The paper has a huge list of references, however I was surprised of not seeing the following: Kieffer, John C., and En-Hui Yang. "Grammar-based codes: a new class of universal lossless source codes." IEEE Transactions on Information Theory 46.3 (2000): 737-754. Carrascosa, Rafael, et al. "The smallest grammar problem as constituents choice and minimal grammar parsing." Algorithms 4.4 (2011): 262-284. Paskov, Hristo S., et al. "Compressive feature learning." Advances in Neural Information Processing Systems. 2013.

Reproducibility: Yes

Additional Feedback:

[Author Response · NeurIPS 2020]

We thank all the reviewers for their time and for their thoughtful comments. We agree with all that was said and will do our best to address it in the final version. Most importantly, we will incorporate the discussion below that aims to comment on a concern raised by reviewer #3 and to answer a question asked by reviewer #1. Moreover, we will add the suggested references, and add a "broader impact" section[1] (We apologize for not realizing that we are required to explicitly say the impact even for a theoretical work like ours.)

**Is grammar-compression useful for vectors and matrices encountered in ML? (a concern raised by reviewer #3)**

We prove that grammar compressions are harder to analyze (without decompression) than simpler ones like RLE. It is natural to wonder: "*Are grammar compressions really more effective than simpler methods like RLE for the data encountered in ML?*" In principle, of course they could be, but are they *actually*? Because, if not, it is not clear why this community should care about the complexity of grammar-compressed linear algebra. Indeed, in many applications, the vectors are very sparse, and RLE is sufficient to get a very strong compression; zip or another, more specialized, grammar-compression could reduce the size further, but the benefit might be negligible.

In the submission, we have only commented on this question in passing. We mentioned that the source file of our paper compresses from 10KB to 4KB with zip, while RLE has negligible effect, and then moved on, supposing that this anecdote extends to many other scenarios that are popular in ML. The reviewer rightfully objects that sequences behave differently from vectors, and therefore this anecdote is not fully convincing.

As the reviewer suggests, it would have been much better to give empirical evidence: showing real-world datasets of vectors and matrices that are of interest to the ML community where grammar compressions make a difference. Fortunately, such a test was already performed in the "Compressed Linear Algebra" paper [3]. In Table 1 the authors present the compression rates achieved with Gzip for five data sets of vectors that are not very sparse (Higgs, Census, Covtype, ImageNet, Mnist8m). For example, the Census [2] data set has sparsity of 0.43 but a much higher compression rate of 0.0584 is achieved with Gzip. We will highlight this reference in the paper, and we will also perform our own experiments with more datasets and include an explicit comparison with RLE. We plan to pick data sets with large vectors that are not sparse containing Boolean, integer or real values, e.g. ISOLET [2]. From a quick test that we performed we see compression rates of about 0.2 even when the sparsity is 0.9.

(True, even in these cases, there may be another scheme that is simpler than zip that achieves strong compression and is easier to compute over; but this is exactly the message of our paper: the "easy solution" of just using any grammar-compression has serious limitations, and the research should be directed towards task-oriented compression schemes. To quote reviewer #4: "*In this sense, proving a limitation of those models will greatly influence future research programmes.*")

**On the weaker result that follows from [1] (an answer for reviewer #1, we will elaborate on this in the paper)**

First, let us remark that the grammar-compressed vector inner product problem is NP-Hard, but we do not think this result is of much practical interest because it is only meaningful when $n$ is logarithmic in $N$, meaning that the vectors are exponentially compressible. On the other hand, our results address the regime where the vectors are only polynomially compressible, and in fact, our result for inner product holds when $n = N^{1/3}$. The previous work [1] had a different, less efficient reduction showing a lower bound for Disjointness. The lower bound that follows is $N^{1/2}$ instead of our $N^{2/3}$, and it only holds for more compressible vectors where $n = N^{1/4}$ rather than our $n = N^{1/3}$. Technically, the novelty is that we manage to encode two sets ($A$ and $B$) into one vector of length $mU$ rather than $m^2U$. This new construction is crucial for the extensions we show in the paper. In particular, we do not see how to prove any lower bound for matrix-vector inner product without building on this new construction.

*We wish to sincerely thank the reviewers and the PC again for their time and help in improving the quality of this work.*

## Footnotes

[1]The section will express that the broader impact is to inform algorithm design for compressed linear algebra, which can lead to faster algorithms for a variety of tasks on large data sets. The ethical consequences depend on the specific application. We do not see any inherently new concerns raised by our results, beyond those that follow from faster algorithms.

# References

[1] A. Abboud, A. Backurs, K. Bringmann, and M. Künnemann. Fine-grained complexity of analyzing compressed data: Quantifying improvements over decompress-and-solve. In *58th IEEE Annual Symposium on Foundations of Computer Science, FOCS 2017, Berkeley, CA, USA, October 15-17, 2017*, pages 192–203, 2017.

[2] D. Dua and C. Graff. UCI machine learning repository, 2017.

[3] A. Elgohary, M. Boehm, P. J. Haas, F. R. Reiss, and B. Reinwald. Compressed linear algebra for declarative large-scale machine learning. *Commun. ACM*, 62(5):83–91, 2019.



[Meta-Review · NeurIPS 2020]

This paper received overall good reviews and is considered novel and of interest. In terms of technical contribution it seems the improvement over previous work ([2]) is somewhat incremental. Another issue that was raised is relevance to the audience. The authors should better explain and justify the connection between their work and the current research performed in ML. Also, perhaps discussing relevant literature in ML on learning algorithms that work over lossless compressed data and how the aforementioned lower bound relates to existing techniques. (see for example, Paskov et al. "Compressive feature learning." 13 and later works)